# Molecular Mutations and Clinical Behavior in Bethesda III and IV Thyroid Nodules: A Comparative Study

**DOI:** 10.3390/cancers16244249

**Published:** 2024-12-20

**Authors:** Alexandra E. Payne, Coralie Lefebvre, Michael Minello, Mohannad Rajab, Sabrina Daniela da Silva, Marc Pusztaszeri, Michael P. Hier, Veronique-Isabelle Forest

**Affiliations:** 1Faculty of Arts and Sciences, Duke University, Durham, NC 27708, USA; 2Faculty of Medicine, McGill University, Montreal, QC H3A 2M7, Canada; coralie.lefebvre@mail.mcgill.ca (C.L.); michael.minello@mail.mcgill.ca (M.M.); 3Department of Otolaryngology Head and Neck Surgery, Sir Mortimer B. Davis-Jewish General Hospital, McGill University, Montreal, QC H3T 1E2, Canada; mmrajab@kfshrc.edu (M.R.); sabrina.wurzba@mcgill.ca (S.D.d.S.); michael.hier.med@ssss.gouv.qc.ca (M.P.H.); veronique-isabe.forest@mcgill.ca (V.-I.F.); 4Department of Otolaryngology—Head and Neck Surgery, King Faisal Specialist Hospital & Research Center, Al Madinah Al Munawwarah 42523, Saudi Arabia; 5Department of Pathology, Sir Mortimer B. Davis-Jewish General Hospital, McGill University, Montreal, QC H3T 1E2, Canada

**Keywords:** thyroid cancer, Bethesda system, indeterminate thyroid nodules, molecular profiling, ThyroSeq v3, clinical decision-making, endocrine malignancy

## Abstract

Thyroid cancer is the most prevalent endocrine malignancy. Thyroid molecular testing is a tool that can help better understand the nature of thyroid tumors. Thyroid nodules are scored on a Bethesda scale from I–VI based on the likelihood of malignancy. Bethesda III and IV nodules are classified as indeterminate; in other words, there is uncertainty as to whether they are benign or malignant. This study analyzes how molecular testing can help to identify specific molecular mutations, copy number alterations, and gene expression profiles within indeterminate thyroid nodules to have a better understanding of them. This will lead to more effective and optimal treatment plans. Molecular testing is a valuable tool that has been well researched with respect to the invasiveness/aggressiveness of nodules and their associations with specific molecular mutations and alterations. Molecular testing was used in this study to evaluate Bethesda III and IV nodules and highlight any patterns, trends, and differences observed between the two groups.

## 1. Introduction

Thyroid cancer is the most prevalent endocrine malignancy. Approximately 5% of the population has a thyroid nodule that can be palpated, yet the likelihood of the nodule being malignant is low [1,2]. The gold standard for accurate diagnosis of thyroid cancer is based on histopathology following surgery. Consequently, the proportion of the population that undergoes diagnostic surgery only to have had benign disease has been reported to be as high as 90% [1]. Thyroid surgery is associated with potential post-surgical complications and also may lead to the need for lifelong thyroid hormone supplementation, which can diminish a patient’s quality of life [3]. Prior to surgical intervention, clinicians utilize ultrasound to image the thyroid nodule followed by a fine needle aspiration (FNA) [1,4]. Cytopathological results of the FNA are classified according to the Bethesda system for reporting thyroid cytopathology (TBSRTC), which ranges from a score of I (non-diagnostic) to VI (malignant) and was endorsed by the American Thyroid Association 2015 guidelines [5,6,7].

The molecular pathogenesis of all types of thyroid carcinoma has been significantly explored in recent years [8]. The majority of thyroid cancer driver alterations lead to dysregulated mitogen-activated protein kinase (MAPK) and phosphatidylinositol-3 kinase (PI3K)-AKT pathways [9]. As a result, genetic analysis of thyroid cancers has become increasingly important. Preoperative molecular testing (MT) of thyroid nodules is a recent, yet highly useful diagnostic tool to characterize thyroid nodules with related molecular markers [10]. ThyroSeq v3 is a MT that identifies a broad spectrum of molecular alterations such as gene fusions, copy number alterations (CNAs), gene expression profiles (GEPs), and point mutations [11]. Thyroid nodules with RAS-like mutations in the MAPK pathway tend to be less proliferative, as they respond to negative feedback inhibition. BRAF-like mutations, however, tend to have much higher levels of cellular proliferation that lead to invasive/aggressive cancer, since they do not respond to negative feedback [9].

Identifying a thyroid nodule’s molecular markers and its clinicopathological features allows for clinicians to personalize treatment plans, optimize patient outcomes, and reduce morbidity [12,13]. In addition, the identification of a molecular alteration can help determine whether a patient requires surgery and, if so, whether they need a hemithyroidectomy or a total thyroidectomy +/− neck dissection. Studies have started to associate Bethesda score to respective molecular alterations. Of note, Bethesda category V and VI thyroid nodules have a strong correlation with *BRAF V600E* mutations, whereas intermediate thyroid nodules categorized as Bethesda III and IV are associated with *H*, *K*, or *NRAS* or EIF1AX mutations, CNAs, or GEP [14]. Likewise, although Bethesda III nodules are more likely to be benign than Bethesda IV, when malignant, they are more likely to have aggressive histopathological features because of their heterogeneity and unpredictability in behavior [12,15]. The risk of malignancy for Bethesda III thyroid nodules is around 5–15% [16,17,18]. The intimate link between Bethesda score and driver mutations, as well as the lack of nuance within the group of intermediate thyroid nodules (Bethesda III and IV), prompts this study’s objective.

This study aims to identify and compare the frequency of molecular alterations in Bethesda III and IV thyroid nodules and to determine if these differences correlate with tumor invasiveness/aggressiveness or malignancy. In conjunction, the goal is to help thyroid specialists manage patients with intermediate thyroid nodules and to determine invasiveness/aggressiveness, optimize patient treatment plans, and minimize diagnostic surgical intervention.

## 2. Materials and Methods

### 2.1. Patient Samples

This is a multicenter retrospective chart review of 2050 patient files who underwent thyroid surgery for their dominant thyroid nodule at one of two McGill University teaching hospitals from January 2016 to April 2022 in Montreal, QC, Canada. The files reviewed were from patients who underwent thyroid surgery between January 2016 and April 2022 for their dominant thyroid nodule. All patients were ≥18 years of age. Patients were included in the study if they demonstrated thyroid nodules categorized as atypia of undetermined significance (AUS/FLUS; Bethesda III) or follicular neoplasm/suspicious of follicular neoplasm (FN/SFN; Bethesda IV) on FNA, had ThyroSeq v3 molecular testing performed on the targeted nodule, and had diagnostic surgery. Ethics approval was obtained by the Research Ethics Committee of the integrated Health and Social Services Network for West-Central Montreal 24 May 2022 (#05-2022-3299).

### 2.2. Data Collection

Data were collected on patient characteristics (including sex and age), tumor characteristics (including ultrasound findings, FNA results, ThyroSeq v3 molecular testing results, and histopathological findings after surgery), and surgical management decisions. There were 316 patient files who underwent ThyroSeq v3 molecular testing and thyroid surgery. A total of 203 patient files qualified to be in the study in adherence with selection criteria. Invasive/aggressive malignancies were observed and defined as having one or more of the following characteristics: lymph node metastasis, poorly differentiated carcinoma, aggressive variants of papillary thyroid cancer (hobnail, tall cell, and columnar), extrathyroidal extensions (ETE) and extensive lymph vascular invasion (LVI) [19,20].

### 2.3. Statistical Analysis

The samples were initially divided into their respective Bethesda group (III and IV) and then into two other groups based on postoperative pathology (Benign/Malignant). Of the tumors that were malignant, they were divided into aggressive and non-aggressive/NIFTP groups. Descriptive analysis was conducted. Associations between Bethesda category, ThyroSeq v3 molecular testing results, age, sex, and histopathology were examined. Statistical significance between groups was defined as *p*-value < 0.05 using a logistic regression model. Statistical analyses of associations between the categorical variables were performed by the chi-square test or two-sided Fisher’s exact test. All analyses were performed using the statistical software package STATA version 18 (STATA Corporation, College Station, TX, USA).

## 3. Results

Of the 316 patient files who underwent ThyroSeq v3 molecular testing at McGill University teaching hospitals between January 2016 and April 2022, 203 thyroid nodules were categorized as either Bethesda III or IV on FNA and underwent diagnostic surgical resection. A total of 117 (57.6%) thyroid nodules were Bethesda III and 86 (42.4%) were Bethesda IV on FNA (Table 1). The average age for the Bethesda III nodules was 54.4 years (ranging from 22 to 88 years old) and 51.9 years (ranging from 16 to 78 years old) for Bethesda IV. Of the Bethesda III thyroid nodules, 90 (76.9%) were female while 27 (23.1%) were male. Of those categorized as Bethesda IV, 73 (84.9%) were female and 13 (15.1) were male. There was no statistical significance reported between Bethesda score and age or gender.

After confirming malignancy on final pathology, 79 (67.5%) Bethesda III nodules and 72 (83.7%) Bethesda IV nodules were malignant/NIFTP (*p* = 0.009). Of those that were malignant, 12 (10.3%) Bethesda III nodules and 11 (12.8%) Bethesda IV nodules demonstrated invasive/aggressive features on final pathology (Table 1).

Of the resected Bethesda III thyroid nodules that were malignant/NIFTP, 65 (82.3%) had a hemithyroidectomy and 14 (17.7%) had a total thyroidectomy. Of the resected Bethesda IV thyroid nodules that were malignant/NIFTP, 65 (90.3%) had a hemithyroidectomy and 7 (9.7%) had a total thyroidectomy (Table 2).

Molecular alterations and mutations detected on ThyroSeq v3 were associated with both Bethesda III and Bethesda IV groups to assess statistical significance (Table 3). There isa statistically significant difference in the presence of CNAs (*p* = 0.008) and the GEP (*p* = 0.005) in malignant Bethesda IV thyroid nodules. Furthermore, despite not meeting the threshold for statistical power defined by this study, there is an observable trend that only malignant Bethesda III nodules demonstrated *BRAF V600E* (n = 4) (*p* = 0.053) mutations on molecular testing, three of which led to aggressive cancer (Table 4).

When looking specifically at the molecular mutations associated, respectively, with aggressive and non-aggressive/NIFTP Bethesda III nodules, we can appreciate some trends (Table 4). An aggressive thyroid nodule is characterized by the presence of extrathyroidal extension, positive lymph nodes, an aggressive variant (hobnail, tall cell, columnar) and being larger than 4 cm in size. Due to limited sample size (<5) or beyond threshold *p*-values (*p* > 0.05) we cannot assess statistical power. However, we noted that *ALK* fusion (8.3%) (*p* = 0.245), *BRAF V600E* (25.0%) (*p* = 0.001 *), and *TP53* (25.0%) (*p* = 0.018 *) tend to associate with invasive/aggressive Bethesda III thyroid nodules. We can also note that all six *THADA* fusions (9.0%) (*p* = 0.117) in Bethesda III nodules were lacking invasive/aggressive features.

In analyzing the aggressive and non-aggressive/NIFTP Bethesda IV thyroid nodules (see Figure 1 and Figure 2), there were no cases with *BRAF V600E* and *KRAS* mutations (Table 5). In contrast, all three *THADA* fusions were in Bethesda IV nodules lacking invasive/aggressive features. The aforementioned observations, however, are noticeable trends, but did not have sufficient sample size to bear statistical power. Furthermore, it was reported that of the nine malignant Bethesda IV nodules with *EIF1AX* mutations, five (55.6%) did not lead to aggressive thyroid cancer.

## 4. Discussion

The Bethesda System for Reporting Thyroid Cytopathology is a well-established tool for estimating the risk of malignancy (ROM) in thyroid nodules [5,21]. Molecular testing for alterations and mutations helps to refine the ROM. Certain mutations, such as *BRAF V600E*, are more likely to be associated with Bethesda V and VI nodules than Bethesda III and IV nodules [22]. Bethesda III and IV nodules are known to behave differently. For example, Bethesda IV nodules are more likely to be malignant or NIFTP than Bethesda III nodules. Interestingly, when Bethesda III nodules are malignant, they are more likely to be more invasive and behave more aggressively when compared to Bethesda IV nodules. Turkdogan et al. demonstrated that among a cohort of malignant thyroid nodules, 18.6% of Bethesda III and 10.2% of Bethesda IV nodules showed aggressive features [15]. Despite these findings, the association between molecular alterations and mutations that may distinguish between Bethesda III and IV nodules is not well defined. The aim of this study was to identify the frequency and nature of molecular mutations and alterations in thyroid nodules classified as Bethesda III and IV and to identify differences that may provide insights into the different clinical behaviors of these indeterminate thyroid nodules.

In this study, Bethesda IV nodules exhibited more frequent copy number alterations and gene expression profile changes than Bethesda III nodules. Interestingly, *EIF1AX* mutations in malignant Bethesda IV thyroid nodules appeared to be associated with more indolent forms of thyroid cancer. Moreover, in this study, all thyroid nodules harboring the *BRAF V600E* mutation were classified as Bethesda III. Consequently, when comparing malignant tumors between Bethesda III and IV categories, Bethesda III cancers exhibited a higher prevalence of the aggressive *BRAF V600E* mutation. In contrast, Bethesda IV tumors were more commonly associated with indolent RAS-like mutations, including *EIF1AX*, along with other molecular alterations such as CNA and GEP [23]. These findings provide a molecular basis for the observed trend of more invasive malignancies in Bethesda III nodules, suggesting that the presence of specific mutations like *BRAF V600E* in these nodules may contribute to their more aggressive clinical behavior.

The existing literature indicates that *RAS*-like mutations are typically associated with non-aggressive thyroid cancers, largely due to their relatively low MAPK signaling activity and a greater responsiveness to negative feedback mechanisms [9]. In contrast, *BRAF*-like mutations, including the *BRAF V600E* mutation, are commonly linked to more aggressive thyroid cancers, characterized by elevated MAPK signaling and a reduced sensitivity to negative feedback [9]. These findings reinforce the notion that while Bethesda III nodules are more likely to be benign, those that are malignant often harbor molecular mutations, such as *BRAF V600E*, which are associated with more aggressive cancer behavior. Conversely, Bethesda IV nodules are more frequently associated with molecular alterations, including *RAS*-like mutations, that tend to lead to less aggressive thyroid cancers. Additionally, the presence of secondary mutations on top of another mutation is commonly associated with higher risk of aggressiveness in thyroid cancer malignancies [24,25].

This study suggests that molecular testing of Bethesda III thyroid nodules may be more important to perform than the testing of Bethesda IV nodules. Although Bethesda III nodules are less likely to be malignant, when they are, they have an increasing risk of invasiveness and clinically aggressive behavior [26,27]. Patients with aggressive mutations such as *BRAF V600E* highly benefit from early detection regarding treatment plans [9,28,29]. Therefore, knowing that Bethesda III patients have an increased risk of aggressivity can encourage earlier diagnoses using molecular testing.

This study has several limitations. First, the retrospective design of the chart review inherently limits the strength of the findings compared to a prospective study design. Additionally, the patient files included in this study were drawn exclusively from McGill University tertiary care centers in Montreal, QC, Canada, introducing a geographic selection bias. Another significant limitation stems from the fact that, at the time of management, some patients were required to pay out-of-pocket for molecular testing, resulting in potential selection bias. Specifically, this financial barrier may have influenced which patients underwent testing, potentially leading to an overrepresentation or underrepresentation of aggressive or non-aggressive molecular mutations and malignancies. To address these limitations, future studies should include a broader and more diverse patient population from multiple centers to reduce selection bias and enhance the generalizability of the findings.

## 5. Conclusions

Bethesda III thyroid nodules were found to have more *BRAF*-like mutations, and Bethesda IV nodules were found to have more *RAS*-like mutations and alterations. This may explain the finding that Bethesda III thyroid nodules, when malignant, behave more aggressively. As a result, molecular testing may in fact be more important to perform in Bethesda III nodules than in Bethesda IV nodules. More studies are required to test the above findings.

This study highlights significant differences in the frequency of molecular mutations between Bethesda III and IV thyroid nodules. Specifically, Bethesda IV nodules exhibited strong associations with CNA and GEP. Notably, when Bethesda IV nodules were malignant, they frequently harbored EIF1AX mutations, which did not typically result in aggressive cancer. In contrast, all malignant tumors with the *BRAF V600E* mutation identified in this study were classified as Bethesda III, suggesting a unique mutation profile for these nodules. Furthermore, the presence of *THADA* fusions was consistently associated with non-aggressive cancers. These findings suggest that while Bethesda III nodules are more often benign, their malignant counterparts are more likely to carry mutations and molecular alterations indicative of aggressive cancer. Understanding these molecular distinctions can aid clinicians in tailoring treatment strategies, thereby enhancing patient care and outcomes.

## Figures and Tables

**Figure 1 cancers-16-04249-f001:**
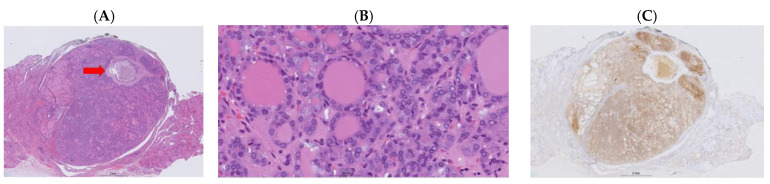
A 46-year-old female with a 1 cm thyroid nodule that was diagnosed as Bethesda IV using FNA. ThyroseqV3 demonstrated the presence of *NRAS Q61R* mutation. Final pathology shows a well demarcated non-invasive follicular neoplasm with papillary-like nuclear features (NIFTP). Post-FNAB changes can be seen in the center of the nodule (**A**, red arrow). Higher magnification shows the follicular architecture and variable nuclear features of papillary thyroid carcinoma (**B**). The tumor was positive for NRAS Q61R using immunohistochemistry (**C**).

**Figure 2 cancers-16-04249-f002:**

A 27-year-old female with a 3 cm thyroid nodule that was diagnosed as Bethesda IV using FNA. ThyroseqV3 demonstrated the presence of *STRN::ALK* fusion. Final pathology was a papillary thyroid carcinoma, classical type, with predominant follicular architecture (**A**,**B**), showing overexpression of ALK using immunohistochemistry (**C**).

**Table 1 cancers-16-04249-t001:** Baseline characteristics of the 203 thyroid nodules classified as Bethesda III and IV.

Variant	Bethesda III (%) (n = 117)	Bethesda IV (%) (n = 86)	*p*-Value
Age (years) mean (min–max, ±SD)	54.4 (22–88, ±12.3)	51.9 (16–78, ±15.0)	0.195
Sex Female Male	90 (76.9) 27 (23.1)	73 (84.9) 13 (15.1)	0.159
Nodule size (cm) mean (min–max, ±SD)	2.17 (0.1–7.5, ±1.4)	2.27 (0.1–6.7, ±1.1)	0.610
Pathology			0.009
Benign	38 (32.5)	14 (16.3)	
Malignant/NIFTP *	79 (67.5)	72 (83.7)	
Aggressive features			0.573
Present	12 (10.3)	11 (12.8)	
Not present/NIFTP	105(89.7)	75 (87.2)	

* NIFTP: Non-invasive follicular thyroid neoplasm with papillary-like nuclear features; SD: standard deviation.

**Table 2 cancers-16-04249-t002:** Pathological and clinical characteristics of patients deemed malignant on final pathology.

Variant	Malignant/NIFTP Bethesda III (%) (n = 79)	Malignant/NIFTP Bethesda IV (%) (n = 72)	*p*-Value
Sex Female Male	60 (76.0) 19 (24.1)	61 (84.7) 11 (15.3)	0.177
Surgical intervention			
Hemithyroidectomy	65 (82.3)	65 (90.3)	0.156
Total Thyroidectomy	14 (17.7)	7 (9.7)	
Type of Malignancy/NIFTP			
NIFTP Papillary carcinoma	11 (13.9) 55 (69.6)	4 (5.6) 54 (75.0)	0.089 0.461
Follicular carcinoma	1 (1.3)	6 (8.3)	0.039
Oncocytic (Hürthle cell) carcinoma	5 (6.3)	3 (4.2)	0.554
Poorly differentiated thyroid carcinoma	1 (1.3)	6 (8.3)	0.039
Well-differentiated follicular neoplasm of undetermined malignant potential	4 (5.1)	3 (4.2)	0.794

**Table 3 cancers-16-04249-t003:** Association between molecular mutation and malignant/NIFTP Bethesda III/IV thyroid nodules.

Mutation	Malignant/NIFTP Bethesda III (%) (n = 79)	Malignant/NIFTP Bethesda IV (%) (n = 72)	*p*-Value
*HRAS*	8 (10.1)	13 (18.1)	0.160
*KRAS*	2 (2.5)	0 (0)	0.174 *
*NRAS*	20 (25.3)	12 (16.7)	0.194
*EIF1AX*	13 (16.5)	9 (12.5)	0.491
*PTEN*	3 (3.8)	3 (4.2)	0.908
*DICER1*	3 (3.8)	8 (11.1)	0.084
*CNA*	11 (13.9)	23 (31.9)	0.008
*GEP*	11 (13.9)	24 (33.3)	0.005
*TP53*	3 (3.8)	3 (4.2)	0.908
*BRAF V600E*	4 (5.1)	0 (0)	0.053 *
*THADA* fusion	6 (7.6)	3 (4.2)	0.374
*ALK* fusion	1 (1.3)	1 (1.4)	0.947 *

* We cannot truly assess statistical power, since the sample size <5. In this case, we can appreciate the trend.

**Table 4 cancers-16-04249-t004:** Association between molecular mutation and aggressive features in Bethesda III thyroid nodules.

Mutation	Non-Aggressive/NIFTP Bethesda III (%) (n = 67)	Aggressive Bethesda III (%) (n = 12)	*p*-Value
*HRAS*	7 (10.4)	1 (8.3)	0.073
*KRAS*	2 (3.0)	0 (0)	0.765 *
*NRAS*	19 (28.4)	1(8.3)	0.033
*EIF1AX*	11 (16.4)	2 (16.7)	0.131
*PTEN*	3 (4.5)	0 (0)	0.712 *
*DICER1*	3 (4.5)	0 (0)	0.130 *
*CNA*	9 (13.4)	2 (16.7)	0.498
*GEP*	10 (14.9)	1 (8.3)	0.431
*TP53*	0 (0)	3 (25.0)	0.018 *
*BRAFV600E*	1 (1.5)	3 (25.0)	0.001 *
*THADA* fusion	6 (9.0)	0 (0)	0.117
*ALK* fusion	0 (0)	1 (8.3)	0.245 *

* We cannot truly assess statistical power, since the sample size <5. In this case, we can appreciate the trend.

**Table 5 cancers-16-04249-t005:** Association between molecular mutation and aggressive features in Bethesda IV thyroid nodules.

Mutation	Non-Aggressive/NIFTP Bethesda IV (%) (n = 61)	Aggressive Bethesda IV (%) (n = 11)	*p*-Value
*HRAS*	11 (18.0)	2 (18.2)	0.855
*KRAS*	0 (0)	0 (0)	- *
*NRAS*	11 (18.0)	1 (9.1)	0.550
*EIF1AX*	5(8.2)	4 (36.4)	0.001
*PTEN*	2 (3.3)	1 (9.1)	0.619 *
*DICER1*	7 (11.5)	1 (9.1)	0.873
*CNA*	20 (32.8)	3 (27.3)	0.689
*GEP*	21 (34.4)	3 (27.3)	0.752
*TP53*	0 (0)	3 (27.3)	0.001 *
*BRAFV600E*	0 (0)	0 (0)	- *
*THADA* fusion	3 (4.9)	0 (0)	0.500 *
*ALK* fusion	0 (0)	1 (9.1)	0.116 *

* We cannot truly assess statistical power, since the sample size <5. In this case, we can appreciate the trend.

## Data Availability

The data presented in this study are available on request from the corresponding author. The data are not publicly available due to the ethics approval agreement.

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
