# Peer review of "Molecular Mutations and Clinical Behavior in Bethesda III and IV Thyroid Nodules: A Comparative Study"

_cancers, 2024, doi:10.3390/cancers16244249_

Round 1

Reviewer 1 Report

Comments and Suggestions for Authors

Payne et al retrospectively investigated the molecular landscape in a cohort of BETHESDA III and IV thyroid nodules .

The article is well written and the methodology used is very appropriate.

I have only minor revisions that could significantly improved the article.

Here some minors points that need to be update:

Materials and methods:

Pag 3 line 96: The authors should clarified the 2050 patients files : they correspond to the dominant nodule for each patient ?

Please update.

Results :

Pag 5 line 162 .

The authors should clarified the definition to the invasive/aggressive BETHESDA III thyroid nodules.

I would suggest to add some ultrasound findings of each nodule (EUTIRADS score, size) and echogenicity of the thyroid parenchyma as well as the presence or not of the capsular invasion.

I recommend also to add the level of thyroid autoantibodies and the TSH in a small table corresponding to the BETHESDA III and IV nodules in order to search if is there is correlation between the most frequent mutation described in table 4 and to the thyroid hormones and antibodies.

I suggest a multivariate analysis.

Author Response

Comment 1: Pag 3 line 96: The authors should clarified the 2050 patients files : they correspond to the dominant nodule for each patient ?

Response 1: Thank you very much for this comment, we appreciate your thorough feedback regarding our findings. You raised an important question about the 2050 patient files and whether they correspond to the dominant nodule for each patient. We have clarified this point on page 3, line 96, to specify that the 2050 patient files indeed correspond to the dominant nodule for each patient.

Comment 2: Pag 5 line 162, The authors should clarified the definition to the invasive/aggressive BETHESDA III thyroid nodules

Response 2: We appreciate your thoughtful feedback regarding our manuscript. You raised an important point about clarifying the definition of invasive/aggressive Bethesda III thyroid nodules. We have addressed this concern by providing a clear definition on page 3, line 113 of the study, and reiterating it on page 6, below Table 5. Your input has been valuable in ensuring that our terminology is precise and consistent throughout the manuscript.

Comment 3: I would suggest to add some ultrasound findings of each nodule (EUTIRADS score, size) and echogenicity of the thyroid parenchyma as well as the presence or not of the capsular invasion.

Response 3: Thank you very much for this comment, we appreciate your thorough feedback regarding our findings. You presented a very interesting point on other markers for the thyroid nodules that could possibly correlate with the mutations. We have found at our institution that the TIARDS score varies depending on the radiologist. Moreover, our patients come from multiple centers, so there is a large discrepancy between them. This would lead to data that is not necessarily helpful in gauging a pattern of drawing meaningful conclusions. Indeed the literature has addressed this point is studies such as “ACR TI-RADS: Pitfalls, Solutions, and Future Directions” (Tappouni et. al, 2019). As such, we decided not to use this as one of our variables after meeting with the statistician. Despite this, we have clarified the markers for aggressive nodules, and tied in some of the factors there which seem to be more representative markers. Nevertheless, we appreciate your valuable insight and hope to investigate this notion in future studies.

Tappouni RR, Itri JN, McQueen TS, Lalwani N, Ou JJ. ACR TI-RADS: Pitfalls, Solutions, and Future Directions. Radiographics. 2019 Nov-Dec;39(7):2040-2052. doi: 10.1148/rg.2019190026. Epub 2019 Oct 11. PMID: 31603734.

Comment 4: 

I recommend also to add the level of thyroid autoantibodies and the TSH in a small table corresponding to the BETHESDA III and IV nodules in order to search if is there is correlation between the most frequent mutation described in table 4 and to the thyroid hormones and antibodies.

I suggest a multivariate analysis.

Response 4: Thank you very much for this comment, it is extremely insightful and provides us with interesting opportunities to consider other factors. However, we do not have datapoints regarding TSH levels or thyroid autoantibodies, as this is outside the scope of our study. While this is a very provocative idea, the focus of this was to observe and analyze mutations present through molecular testing in the different Bethesda III and IV thyroid nodules respectively. This would be an extremely interesting subject to explore in a future study, and could provide further insight into understanding these nodules and how to approach them. Interestingly it has given us a new idea to investigate whether TSH levels and thyroid autoantibodies. We hope to do this study in the near future.

Reviewer 2 Report

Comments and Suggestions for Authors

In this study, the authors studied molecular changes and clinical behavior in Bethesda group III and IV thyroid nodules. I have the following comments and questions.

1. Although patient cohort is from a teaching hospital of McGill University, two authors (first author and fourth author) are not affiliated with McGill University. Clarification is needed regarding the contributions of these two authors to the study.

2. The risk of malignancy (ROM) for Bethesda categories III and IV in this study is significantly higher than the values reported in the Bethesda guidelines. Specifically, the ROM for categories III and IV are 67.5% and 83.7%, respectively, compared to the average ROM of 16% and 23% outlined in the guidelines. What accounts for this substantial discrepancy? Re-evaluating the cytology specimens with pathologists is recommended.

3. The majority of malignancies in both the III and IV groups are PTC. However, only a small percentage of cases harbored the BRAF V600E mutation. Including PTC subtyping in the analysis is necessary for a more understanding of the data.

4. What criteria are used to define the aggressive features of malignancy?

5. Include representative histological images for both cytology and surgical resections.

6. Of the 11 references cited, 4 appear to originate from the same institution and involve overlapping authorship with the current study. To minimize self-citation, it is advisable to explore additional references.

Author Response

Comment 1: 1. Although patient cohort is from a teaching hospital of McGill University, two authors (first author and fourth author) are not affiliated with McGill University. Clarification is needed regarding the contributions of these two authors to the study.

Response 1: We appreciate this comment, thank you very much for the insight. Author number 4, Mohannad Rajab, was a fellow in Otolaryngology at McGill University teaching hospitals in Montreal, where he practiced for one year. He contributed greatly to the findings of this paper and to patient care. The first author, Alexandra Payne, has conducted many research studies with the team from McGill University, and began this study in the summer while working at the ENT-Otolaryngology clinic in Montreal, with the team from McGill. We hope that this clarifies any confusion that might have arisen.

Comment 2: The risk of malignancy (ROM) for Bethesda categories III and IV in this study is significantly higher than the values reported in the Bethesda guidelines. Specifically, the ROM for categories III and IV are 67.5% and 83.7%, respectively, compared to the average ROM of 16% and 23% outlined in the guidelines. What accounts for this substantial discrepancy? Re-evaluating the cytology specimens with pathologists is recommended.

Response 2: This is a very perceptive comment, thank you very much for your insight. All of the patients that were included in this study underwent both molecular testing and surgery. As such, this sample of patients is not representative of patients included in the average ROM statistics. These patients were selected for surgery because of a range of criteria. The ROM statistics presented here include all patients, many of whom did not have surgery and who did not require it. As such, these statistics are not representative of the patient sample we used. Additionally, all the patients included had positive molecular tests which indicated a need for surgery, adding another layer of severity compared to the average ROM data.

Comment 3: The majority of malignancies in both the III and IV groups are PTC. However, only a small percentage of cases harbored the BRAF V600E mutation. Including PTC subtyping in the analysis is necessary for a more understanding of the data.

Response 3: Thank you for this remark, we value your feedback greatly. Indeed the majority of malignancies for Bethesda III and IV nodules are PTC, yet only a small subtype of the cases harbored a BRAF V600E mutation. In essence, the majority of Bethesda V and VI harbour BRAF mutations, however Bethesda III and IV nodules are less likely to harbour this mutation. In addition, mutation identification of the nodules that were not BRAFV600E is included in the Results section in Table 3 on page 6. Thank you very much for your feedback and grasp of the subject matter.

Comment 4: What criteria are used to define the aggressive features of malignancy?

Response 4: This is a very discerning comment, thank you very much for the critique. This is an essential point of clarification for the study, so we appreciate the feedback. We have added clarity as to what defines an aggressive feature of malignancy. This can be found on page 3 line 113, as well as an page 6 line 172, where the characteristics are clearly stated. Thank you once again for this insight.

Comment 5: Include representative histological images for both cytology and surgical resections.

Response 5: Thank you very much for this recommendation, we have indeed added histological images for the cytology and surgical resections. This can be found in the results section under “Figure 1” and “Figure 2” on pages 8 and 9 respectively. We hope this provides further clarity and depth to our study.

Comment 6: Of the 11 references cited, 4 appear to originate from the same institution and involve overlapping authorship with the current study. To minimize self-citation, it is advisable to explore additional references.

Response 6: Thank you very much for this comment, it is much appreciated. We have added many additional references from different institutions and authors in order to minimize self-citation. Thank you once again for this observation.

Round 2

Reviewer 2 Report

Comments and Suggestions for Authors

I don't have any comments. Thank you for answering my questions.

Comments on the Quality of English Language

English is fine. No comments.